# Association between Emotional Eating, Depressive Symptoms and Laryngopharyngeal Reflux Symptoms in College Students: A Cross-Sectional Study in Hunan

**DOI:** 10.3390/nu12061595

**Published:** 2020-05-29

**Authors:** Hanmei Liu, Qiping Yang, Jing Luo, Yufeng Ouyang, Minghui Sun, Yue Xi, Cuiting Yong, Caihong Xiang, Qian Lin

**Affiliations:** Department of Nutrition Science and Food Hygiene, Xiangya School of Public Health, Central South University, 110 Xiangya Rd, Changsha 410078, China; hanmeiliu@csu.edu.cn (H.L.); yangqiping12@csu.edu.cn (Q.Y.); luojing2546@csu.edu.cn (J.L.); oyyf0102@csu.edu.cn (Y.O.); sun.1234@csu.edu.cn (M.S.); summerxi@csu.edu.cn (Y.X.); yongcuiting@csu.edu.cn (C.Y.); xch0622@csu.edu.cn (C.X.)

**Keywords:** emotional eating, depressive symptoms, laryngopharyngeal reflux

## Abstract

This study aims to explore associations between emotional eating, depression and laryngopharyngeal reflux among college students in Hunan Province. Methods: This cross-sectional study was conducted among 1301 students at two universities in Hunan. Electronic questionnaires were used to collect information about the students’ emotional eating, depressive symptoms, laryngopharyngeal reflux and sociodemographic characteristics. Anthropometric measurements were collected to obtain body mass index (BMI). Results: High emotional eating was reported by 52.7% of students. The prevalence of depressive symptoms was 18.6% and that of laryngopharyngeal reflux symptoms 8.1%. Both emotional eating and depressive symptoms were associated with laryngopharyngeal reflux symptoms (AOR = 3.822, 95% CI 2.126–6.871 vs. AOR = 4.093, 95% CI 2.516–6.661). Conclusion: The prevalence of emotional eating and depressive symptoms among Chinese college students should be pay more attention in the future. Emotional eating and depressive symptoms were positively associated with laryngopharyngeal symptoms. The characteristics of emotional eating require further study so that effective interventions to promote laryngopharyngeal health among college students may be formulated.

## 1. Introduction

Emotional eating (EE) has been defined as “the tendency to eat in response to a range of negative emotions such as anxiety, depression, anger and loneliness, to cope with negative affect” [1]. In a survey conducted in Finland, 25%–30% of people reported choosing to eat in response to stress [2]. Over the past few decades, the number of individuals with EE has increased significantly [3] and EE is associated with an increase of negative emotions such as depression, stress and anxiety in the population. In China, approximately 23.8% of first-year college students have depressive symptoms [4], The prevalence of depressive symptoms is increasing [5]. With increasing levels of various negative emotions, the number of individuals with EE will gradually increase. Dietary changes caused by negative emotions and emotional eating are often increased intake of high energy density foods, such as sugary foods, sweets or fried foods, etc. [6], which may lead to obesity and increase the risk of chronic diseases [7]. Emotional eating has been linked to overeating and/or fast-eating [8]. These eating habits bring extra burdens on the digestive system. However, few studies have clarified the effects of depression or EE on digestive tract function.

Digestive function is closely related to a daily diet. Studies have shown that continuous eating and anorexia nervosa, binge-eating–purging type can increase the risk of acid reflux [9,10]. Negative emotions and EE may increase the risk of eating disorders such as overeating and binge-eating [11]. Laryngopharyngeal reflux (LPR), a type of gastric acid reflux, is described as retrograde reflux of gastro-duodenal contents into the larynx and pharynx, leading to severe damage of the upper aerodigestive tract [12]. About 5% of people in the Chinese population have LPR [13]. LPR can cause severe head and neck diseases such as hoarseness, taste damage, tooth erosion and cancer [14,15,16,17,18]. To date, the pathogenesis of LPR has not been wholly elucidated [19]. Depression, alcoholism, fermented food, and overeating may increase the risk of LPR [20,21,22]. It is well known that EE behavior can lead to overeating or eating disorders [8]. Studies have also shown that EE was associated with high intakes of high-density snack foods [6]. High fat consumption was associated with an increased risk of gastroesophageal reflux disease (GERD) symptoms [23]. Hence, we are wondering: Is LPR, which similar to GERD, also related to dietary intake? Is emotional eating a risk factor for LPR?

Although studies have explored the relationship between dietary intake and GERD [23,24,25], few have investigated the association between diet and LPR symptoms. The damage of negative emotions and EE on the upper aerodigestive tract function is unknown. LPR and gastroesophageal reflux are both acid refluxes. The response of the larynx differs from that of the esophagus in acid reflux. Because the larynx lacks defense mechanisms to protect against damage by refluxate present in the throat, chronic laryngeal inflammation caused by LPR may trigger neoplastic lesion [26]. To date, the risk factors of LPR symptoms have not been completely elucidated. If diet or psychological factors related to LPR can be identified, and appropriate interventions made, LPR disease may be prevented. Therefore, this study aims to investigate both the prevalence of LPR and the relationship between depressive symptoms, EE and LPR symptoms among first-year college students. It will provide a reference for interventions of EE and depression and contribute to the understanding of LPR.

## 2. Methods

### 2.1. Study Design and Participants

In this cross-sectional study, participants were recruited from two universities in Changsha, Hunan Province, China between May and September 2019. Recruitment was targeted at first-year students and second-year students. The inclusion criteria for participants are summarized as follows: (1) undergraduate students enrolled in 2017 or 2018; (2) capable of providing informed consent and willingness to participate in the study; (3) able to read and fully comprehend the questionnaires items well. Exclusion criteria: current active infection or acute illness of any kind.

No study has investigated the prevalence of EE in Chinese college students. According to the results of similar surveys in other countries, approximately 39% of adults are high emotional eaters (prevalence among man is lower than among woman) [6]. The required sample size was 1047, as calculated using PASS software (version 11.0 for Windows; NCSS LLC, Kaysville, UT, USA), with an expected prevalence of high emotional eaters of 31.6% and an allowable error of 3%. A total of 1307 students were recruited; students who did not complete the questionnaires or unreliable questionnaires (for example, frequencies of all the food categories were reported as “never”), resulting in a sample of 1301 participants. The study was approved by the Ethics Review Committee of the Xiangya School of Public Health, Central South University (No. XYGW-2019-026). Electronic informed consent was obtained from each participant.

### 2.2. Measures

#### 2.2.1. Questionnaire Survey

We used cluster sampling to collect samples from Central South University and Hunan Normal University. Data were collected through Questionnaire Star in a class room situation after informed consent was obtained. The Questionnaire Star is a tool used to develop electronic questionnaires. Each questionnaire had a unique QR code, that could be scanned by the WeChat or QQ app on a smartphone [27]. Two methods were employed to collect samples: (1) asking lecturers to inform students about the survey; and (2) posting posters on campus and sharing copies with WeChat and QQ groups to students to participate at the designated time and place. In addition to demographics information, EE, depressive symptoms, physical activity, LPR symptoms, and food consumption were assessed by the way.

Demographic characteristics: Each student’s sex, age, ethnicity, sibling status (only child, (yes vs. no), place of residence, academic major and monthly living expenses.

Emotional eating: Emotional eating refers to the tendency to eat in a negative emotional state [28]. Emotional eating was assessed using the Chinese version of three-factor eating questionnaire (TFEQ-R18V3) [28], which revised by TFEQ-R21 among Chinese college students [29].The revised Chinese version has good reliability and validity among college students [28]. The TFEQ-R18 uses a 4-point response scale for 18 items, 6 of measure EE, with a score ranging from 6 to 24. Cronbach’s alpha for the EE construct was 0.919. The raw scale scores are transformed to a 0–100 scale ((raw score−lowest possible raw score)/possible raw score range) × 100. Higher scores in the respective scales are indicative of more intense EE. This study referred to the classification method of Camilleri et al., and EE scores were categorized based on the sex-specific median cut-points (excluding those with no EE): (1) no EE (score = 0); (2) low EE (0 < score < median); and (3) high EE (score ≥ median) [6].

Depressive symptoms: Patient Health Questionnaire-9 (PHQ-9) was used to assess depressive symptoms [30]. The PHQ-9 consists of 9 items on a 4-point response scale (from 0, never, to 3, almost every day). PHQ-9 score ranges from 0 to 27, with a lower score corresponding to fewer depressive symptoms. Scoring is classified as <10 as having no depressive symptoms, ≥10 representing depressive symptoms [30]. The Cronbach α reliability coefficient of the items was 0.887 in this study.

Physical activity: The short form self-administered instruments of the international physical activity questionnaire (IPAQ) were used to assess physical activity [31]. We used the instructions provided in the IPAQ manual for reliability and validity, which are detailed elsewhere [32]. We used an overall index of metabolic equivalent (MET, min/week) to present the intensity of physical activity. We used the recommended categorical scoring, three levels of PA (low, moderate and high) as proposed in the IPAQ Scoring Protocol (short form) [32]. Physical activity levels were divided into three grades: low activity or inactivity, moderate physical activity and high physical activity.

Laryngopharyngeal reflux symptoms: Symptoms were assessed using the reflux symptom index (RSI) [33,34], which is getting increasingly recognized by otolaryngologists [35]. The RSI consists of 9 items on a 6-point response scale (from 0, No problem, to 5, severe problem). RSI scores range from 0 to 45; the higher the RSI score, the more severe the LPR symptoms. Scoring is classified as 0–13 as having no LPR symptoms or >13 representing LPR symptoms [33]. Cronbach’s alpha of this test was 0.910.

Food consumption: We used the food frequency questionnaires (FFQ) with 32 food items to measure food consumption during the previous 12 months. According to the dietary characteristics of college students, the FFQ was modified on the basis of Weng et al. [36]. The consumption frequency of sweet and/or fatty foods and salty fatty foods were calculated. Sweet and/or fatty foods include cake, cookies and chocolate, while salty fatty foods include fried chicken, hamburgers and processed meat. Consumption frequencies were categorized by ≥3 times/week or <3 times/week.

#### 2.2.2. Anthropometric Measurements

The height and weight of each participant were measured by TANITA human body composition analyzer BC-W02C (Guangdong Food and Drug Administration (prospective), no. 2210704, 2014), to calculate body mass index (BMI kg/m^2^). According to the BMI standard for Chinese adults, the healthy BMI range is 18.5–23.9 kg/m^2^. Those lower than the healthy range are underweight, those 24.0–27.9 kg/m^2^ are overweight and those higher than 28.0 kg/m^2^ are obese.

### 2.3. Statistical Analysis

SPSS 18.0 software (IBM Corp., Armonk, NY, USA) was used for statistical descriptions, chi-squared tests and logistic regression analyses. The chi-squared test was used to assess differences in the EE and depressive symptoms (with demographic data) and to assess the relationship between EE and, depressive symptoms and LPR symptoms. The participants were classified into LPR symptoms (RSI score > 13) and no LPR symptom (RSI score 0–13). Multiple logistic regression was performed for two models, to examine the variables associated with LPR symptoms, and their 95% confidence intervals (CIs) were computed from these models. Odds ratios (OR) were estimated, adjusting for sex, age, body mass index, physical activity, major, depressive symptoms and emotional eating (Model 1); adjusting for Model 1 + sweet and/or fatty foods + salty fatty foods (Model 2). The prevalence of EE, depressive symptoms, and LPR symptoms are expressed as a percentage of N (%). The significance level was set at *p* < 0.05.

## 3. Results

### 3.1. Characteristics of Participants

A total of 1301 undergraduate students participated in this survey. Of these, 61.0% were female. Ages ranged from 17 to 25 years (mean = 19.8; SD = 0.9) and a plurality (46.6%) were 19 years old. Mean BMI was 21.4 kg/m^2^ (SD = 3.9), which was higher among men (22.5 ± 4.0 kg/m^2^) than females 20.8 ± 3.6 (20.8 ± 3.6 kg/m^2^, *p* < 0.001). In all, 45.7% of the students were an only child and 53.3% were registered rural residents. Overall, 3.5% were obese, 11.6% were overweight and 16.1% were underweight. The median score of overall MET score was 1386.0 min/week. Less than a fifth of the participants (17.1%) were physically active and 25.0% were physically inactive. Fewer males than females were emotional eaters (high EE: 48.4% vs. 55.4%; *p* < 0.001). The mean RSI score was 3.69 (SD = 6.1). The prevalence of depressive and LPR symptoms among the students was 18.6% and 8.1%, respectively (Table 1 and Table 2). Regarding the intake of sweet and/or fatty foods and salty fatty foods, 10.1% consumed sweet and/or fatty foods more than three times per week and 6.6% ate more than salty fatty foods per week (Table 3).

Emotional eating scores varied significantly by major, physical activity levels, BMI, depressive symptoms and LPR symptoms. Females (median = 38.89, 27.78–61.11) had higher EE scores than males (median = 27.78, 5.56–38.89) (*p* < 0.001) (Figure 1). College students whose major was a liberal art (*p* < 0.05), with higher BMI (*p* < 0.01) and with low physical activity levels (*p* < 0.01) had higher EE scores. Of the those with depressive symptoms, 67.8% had higher EE scores, which was significantly higher than for those who did not have report depressive symptoms (49.2%) (*p* < 0.001) (Table 1).

### 3.2. Association between LPR Symptoms, Food Intake, Emotional Eating and Depressive Symptoms

In this study, the prevalence of LPR symptoms was 8.1% in college students. Individuals reporting high EE had higher scores for LPR symptoms than those with no EE or low EE (12.4% vs. 3.2%, *p* < 0.001). Students with depressive symptoms showed more LPR symptoms, compared those without depressive symptoms (21.5% vs. 5.0%, *p* < 0.001). Compared to low consumption, both high consumption of sweet and/or fatty foods (24.2% vs. 6.2%, *p* < 0.001) and high consumption of salty fatty foods (24.4% vs. 6.9%, *p* < 0.001) were associated with higher LPR symptoms (Table 2).

We also observed the association between dietary intake and emotional eating among college students. Among both boys and girls, high EE was associated with the consumption of sweet and/or fatty foods (males: 17.1% vs. 6.5%, *p* < 0.001; females: 11.8% vs. 5.9%, *p* = 0.004). Only for females, high EE was associated with the intake of salty fatty foods (5.9% vs. 2.8% vs. 11.5%, *p* = 0.005). For both sexes, depressive symptoms were associated with higher consumption of both sweet and/or fatty foods (17.4% vs. 8.5%, *p* < 0.001) and salty fatty foods (12.8% vs. 5.2%, *p* < 0.001) (Table 3).

Multivariate logistic regression models were used to analyze the relationships between LPR symptoms as the dependent variable, EE and depressive symptoms. In model 1, compared with college students with no EE, those with high EE were more likely to have higher LPR symptoms scores (AOR = 4.219, 95% CI 1.273–13.977). Students with depressive symptoms were more likely to have higher LPR symptoms scores than those without depressive symptoms (AOR = 3.600, 95% CI 2.159–6.002). Compared with students with low physical activity, those with moderate physical activity were less likely to report LPR symptoms (AOR = 0.302, 95% CI 0.173–0.528). These associations remained after adjustment for sweet and/or fatty foods. Higher intake of sweet and/or fatty foods was related to higher LPR symptom score (AOR = 3.910, 95% CI 2.168–7.051). There are no significant association between the intake of salty fatty foods and LPR symptom score (Table 4).

## 4. Discussion

The purpose of our study was to investigate the prevalence of depressive symptoms, emotional eating and LPR symptoms among first-year college students and to explore the relationship between depressive symptoms, EE and LPR symptoms. Our results indicate that the prevalence of LPR symptoms was 8.1% (RSI > 13), which is higher than the investigation in other regions of China (3.1%–5.0%) [13]. Compared with other countries, the prevalence of LPR was higher than in Japan (7.1%) [37] and it was lower than in Greece (18.8%) [38,39]. We used the RSI scale to investigate LPR symptoms. Individuals may experience LPR symptoms, but not yet have reached the diagnostic criteria for LPR disease. A survey found that in the United States, LPR has become a common adult chronic disease [14]. We should pay closer attention to early symptoms associated with LPR, so that we can adopt appropriate interventions and reduce the risk of chronic disease. Compared with patients with GERD patients, those with LPR have different symptoms and pathophysiological mechanisms. Most patients with LPR do not have esophagitis or heartburn, but feel discomfort in larynx area [26], so it is not always easy to associate LPR symptoms with dietary habits. In addition, because the larynx lacks the epithelial defense mechanism available to esophagus, protecting itself from acid attacks is difficult [26], which may result in severe head and neck diseases [14,15,16,17,18]. Therefore, greater attention should be paid to LPR. Many studies have uncovered that certain dietary habits are related to LPR, such as fermented foods [20] and excessive drinking [22]. In addition, some studies have suggested that certain diets, such as low-acid, and plant protein diets have positive effects in the treatments of LPR [15,40]. Studies have also found that low-fat diets are helpful in the treatment of esophageal reflux disease [41]. In this study, we determined that the frequency of high-sweet/high-fat intake was positively associated with LPR. Therefore, we suspect that reducing the intake of high-energy foods can prevent LPR disease. What is more, moderate physical activity was benefit for reducing of LPR symptoms.

Laryngopharyngeal reflux was a controversial topic due to inconsistencies in epidemiological and etiological data [42]. Previous studies have shown that certain eating behaviors, such as the intake of carbonated beverages, meals and chocolate may damage esophageal function or trigger gastric acid secretion [43,44] However, few studies have explored the association between negative emotions and EE and LPR symptoms. In the present study, we found that depressive symptoms and EE was positively associated with LPR symptoms. Emotional eating refers to a tendency to eat when negative emotions are present. High-level EE (i.e., the group with a high EE score) was likely to reflect two performances when individual eats. First is, the propensity to overeat or binge-eating when experiencing negative emotions [8]. Studies have demonstrated that overeating behavior, such as eat-all-the-time or eat-too-full was a risk factor for LPR and acid reflux [10,45]. Acid secretion increases with increased gastric content. Serious eating disorders, such as binge-eating with self-induced vomiting, may risk incidence of acid reflux [18,46]. Eventually, gastric acid attacks the larynx and triggers various LPR symptoms. Second, the food consumption of emotional eaters differs from others. In our study, individuals with high EE scores consumed high-energy foods more frequently, and high-energy foods were positively associated with LPR symptoms. Few studies have reported the relationship between food consumption and LPR disease, but one study suggested that high dietary fat intake is associated with an increased risk of GERD symptoms and erosive esophagitis [23].

Negative emotions and depressions are related to various digestive diseases [47]. Similar to the findings of other studies, our findings signified an association between depressive symptoms and LPR symptoms [48]. However, many studies have found that individuals with gastrointestinal disorders were more likely to be depressed [49]. Longitudinal studies would be valuable in clarifying the causal relationship between depression and LPR. According to our results, depression and EE are related to LPR symptoms. LPR can increase depressive symptoms and then negative emotions aggravate LPR symptoms by changing dietary behaviors. As a result, a vicious circle may form. Further research is needed to confirm these results. In summary, the association between LPR and dietary behaviors cannot be ignored. Whether with patients with LPR individuals with LPR symptoms, detection as early as possible is crucial, EE may be effectively limited by appropriate interventions, such as the mindfulness [50], physical activity [51] or other intervention methods [52].

The prevalence of depressive symptoms in our study was 18.6%, slightly higher than was reported by Feng et al. [53]—and similar to the results of a systematic review in China [54]—but it was lower than in other countries (30.6%) [55]. Although college students are able to acquire a better education, they have a higher risk of depression than other groups around the world [55,56,57,58]. With increasing depression, EE is increasing [58]. Emotional eating has increased significantly in the past 20 years [3]. In our study, numerous students had high EE scores. In our study, we used the median (0–100) to represent emotional eating score, and the results were similar to the adults in other countries [59,60], indicating that EE was high among these students. However, some studies use mean values to represent EE scores [61,62,63]. It is necessary to inform the EE level of a unified evaluation standard for comparison. Psychological factors affect eating behaviors, and both are determinants of physical health. In China, people are beginning more attention to eating behaviors and psychological problems. We require additional evidence to clarify the associations between negative emotions, unhealthy eating behaviors and LPR symptoms.

No differences in depressive symptoms for sex or academic major were identified. However, lower monthly living expenses, low physical activity and high physical activity were positively associated with depressive symptoms, which is similar to other studies [64]. Socioeconomic conditions are a common influencing factor for depression. Reasonable exercise frequency is beneficial for the reduction of negative emotions in college students. Heavier people had higher EE scores, as in other studies [65]. However, the causal relationship between EE and overweight and obesity is unclear. Longitudinal studies have suggested that EE may increase weight [57,58]. Weight gain can also cause depression [65], which may lead to higher levels of EE [66]. Therefore, high emotional eaters should be identified as early as possible so that they can be guided to form a healthy diet, which can effectively control obesity and reduce the incidence of chronic diseases. In this study, medical students and students with higher physical activity reported lower EE scores, perhaps because they were more health-conscious than others [67,68]. Improved health education in early life may prevent college students from developing EE when they are older.

We recruited a sufficient sample. Moreover, the tool of this investigation was electronic questionnaire, and participants had to complete it before submitting it, which ensured the integrity of the questionnaire data. For the first time, we identified an association between depressive symptoms and EE with LPR symptoms. However, the study had limitations. First, the population in our study cannot reflect the prevalence of EE, depressive and LPR symptoms among all Chinese college students. Second, the cross-sectional design of the study does not allow to infer any direction of causality among the studied variables. Third, although we used the TFEQ-R18 V3 scale to quantify EE, we did not know the specific number of occurrences of EE in the prior period. In other words, even if a participant had a high EE score, it is uneasy to determine whether their food consumption changed if they had not experienced negative emotions. Fourth, this study does not include quantitative data of dietary intake (e.g., calories, sugar and fat). Future study should include the information of period of emotional eating and dietary survey, for a better understanding of dietary intake of people with emotional eating. At the same time, a prospective study of a large population should be conducted to clarify the association between EE and LPR or other diseases. The results of this study can provide a rough picture of the prevalence of EE, depressive symptoms and LPR symptoms among college students in China and can help improve student’s health perceptions and dietary behaviors. More important, it may provide a reference for the guidance of LPR interventions.

## 5. Conclusions

In this study, we found emotional eating and depressive symptoms were associated with LPR symptoms. The prevalence of high EE, depressive symptoms, and LPR symptoms were 52.6%, 18.6% and 8.1% among college students; We should adopt reasonable measures to reduce emotional eating, unhealthy eating habits and depression to prevent LPR disease.

## Figures and Tables

**Figure 1 nutrients-12-01595-f001:**
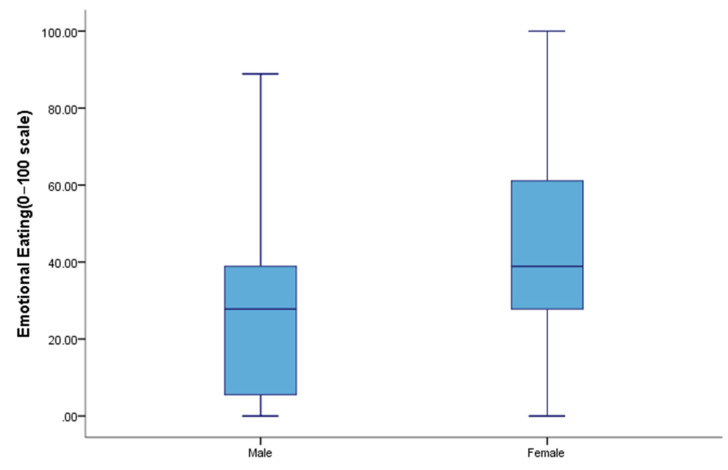
Emotional eating scores by sex.

**Table 1 nutrients-12-01595-t001:** Demographic characteristics of the participants and prevalence of high emotional eating (TFEQ-R18V3 ≥ median) and depressive symptoms (PHQ-9 ≥ 10).

		N (%)	High Emotional Eating, *n* (%)	*P*	Depressive Symptoms, *n* (%)	*P*
Total		1301	685 (52.7%)		242 (18.6%)	
Sex	Male	508 (39.0%)	246 (48.4%)	0.015	107 (21.1%)	0.068
	Female	793 (61.0%)	439 (55.4%)		135 (17.0%)	
Age	≤18 years	195 (15.2%)	99 (50.8%)	0.100	34 (17.4%)	0.943
	19 years	598 (46.6%)	334 (55.8%)		112 (18.7%)	
	20 years	373 (29.0%)	194 (52.0%)		73 (19.6%)	
	≥21 years	118 (9.2%)	52 (44.1%)		22 (18.6%)	
Only child	Yes	595 (45.7%)	315 (52.9%)	0.848	106 (17.8%)	0.504
	No	706 (54.3%)	370 (52.4%)		136 (19.3%)	
Registered residence	Urban	694 (53.3%)	372 (53.6%)	0.463	142 (20.5%)	0.065
	Rural	607 (46.7%)	313 (51.6%)		100 (16.5%)	
Academic major	Liberal arts	179 (13.8%)	108 (60.3%)	0.025	33 (18.4%)	0.693
	Science and technology	840 (64.6%)	442 (52.6%)		163 (19.4%)	
	Medicine	160 (12.3%)	70 (43.8%)		25 (15.6%)	
	Other	122 (9.4%)	65 (53.3%)		21 (17.2%)	
Monthly expenditure	≤500 RMB ^1^	20 (1.6%)	7 (35.0%)	0.377	11 (55.0%)	<0.001
	500–1000 RMB	265 (20.7%)	139 (52.5%)		48 (18.1%)	
	1000–2000 RMB	897 (69.9%)	475 (53.0%)		162 (18.1%)	
	>2000 RMB	101 (7.9%)	57 (56.4%)		18 (17.8%)	
BMI	Underweight	158 (16.1%)	68 (43.0%)	0.007	20 (12.7%)	0.101
	Normal	673 (68.7%)	364 (54.1%)		123 (18.3%)	
	Overweight	114 (11.6%)	69 (60.5%)		19 (16.7%)	
	Obese	34 (3.5%)	23 (67.6%)		10 (29.4%)	
Physical activity	Low	325 (25.0%)	193 (59.4%)	0.002	53 (23.8%)	<0.001
	Moderate	753 (57.9%)	393 (52.2%)		111 (14.7%)	
	High	223 (17.1%)	99 (44.4%)		78 (24.0%)	
Depressive symptoms	Yes	242 (18.6%)	164 (67.8%)	<0.001	–	
	No	1059 (81.5%)	521 (49.2%)		–	

TFEQ-R18V3: three-factor eating questionnaire; PHQ-9: Patient Health Questionnaire-9; BMI: body mass index; RMB: Renminbi (Chinese currency, ^1^ RMB = 0.14 USD). Chi-squared test was performed to compare the groups.

**Table 2 nutrients-12-01595-t002:** Associations between LPR symptoms and food intake, emotional eating and depressive symptoms (reflux symptom index (RSI) > 13).

	LPR Symptoms			
	N	*n* (%)	χ^2^	*P*
Sweet and/or fatty foods			51.782	<0.001
<3 times/week	1169	73 (6.2%)		
≥3 times/week	132	32 (24.2%)		
Salty fatty foods			33.171	<0.001
<3 times/week	1215	84 (6.9%)		
≥3 times/week	86	21 (24.4%)		
Emotional eating			36.695	<0.001
No/Low EE	616	20 (3.2%)		
High EE	685	85 (12.4%)		
Depressive symptoms			72.133	<0.001
Yes	242	52 (21.5%)		
No	1059	53 (5.0%)		
Total	1301	105 (8.1%)		

LPR symptoms: laryngopharyngeal reflux symptoms; EE: emotional eating; Chi-squared test was performed to compare the groups.

**Table 3 nutrients-12-01595-t003:** Dietary intake and emotional eating in college students.

		N	Sweet and/or Fatty Foods ≥3 Times/Week	χ^2^	*P*	Salty Fatty Foods ≥3 Times/Week	χ^2^	*P*
Emotional eating								
Boys	No EE/low EE	262	17 (6.5%)	13.847	<0.001	23 (8.8%)	0.690	0.406
	High EE	246	42 (17.1%)			27 (11.0%)		
	Total	508	59 (11.6%)			50 (9.8%)		
Girls	No EE/low EE	51	21 (5.9%)	8.198	0.004	10 (2.8%)	4.339	0.037
	High EE	439	52 (11.8%)			26 (5.9%)		
	Total	793	73 (9.2%)			36 (4.5%)		
Depressive symptoms				16.949	<0.001		18.519	<0.001
	Yes	242	42 (17.4%)			31 (12.8%)		
	No	1059	90 (8.5%)			55 (5.2%)		
	Total	1301	132 (10.1%)			86 (6.6%)		

EE: emotional eating; Chi-squared test was performed to compare the groups.

**Table 4 nutrients-12-01595-t004:** Logistic regression analysis of the related factors of LPR symptoms (OR (95% CI)).

	Crude OR (95% CI)	Model 1 (OR (95% CI))	Model 2 (OR (95% CI))
Physical activity (reference = Low)			
Moderate	0.290 (0.184, 0.456) ***	0.302 (0.173, 0.528) ***	0.345 (0.195, 0.612) ***
High	3.600 (0.348, 1.034)	0.707 (0.356, 1.404)	0.789 (0.391, 1.593)
Depressive symptoms	5.195 (3.438, 7.850) ***	3.600 (2.159, 6.002) ***	3.439 (2.034, 5.814) ***
Emotional eating (reference = No EE)			
Low EE	1.805 (0.522, 6.249)	1.293 (0.352, 4.745)	1.394 (0.376, 5.169)
High EE	6.800 (2.120, 21.815) ***	4.219 (1.273, 13.977) *	3.934 (1.175, 13.168) *
Sweet and/or fatty foods (reference <3 times/week)	4.804 (3.023, 7.635) ***		3.910 (2.168, 7.051) ***

Model 1: adjusted for sex, age, BMI, physical activity, academic major, depressive symptoms and EE;.Model 2: model 1 + sweet and/or fatty foods + Salty fatty foods; OR: odd ratio; CI: confidence interval; LPR: laryngopharyngeal reflux; EE: emotional eating; *** *p* < 0.001, * *p* < 0.05.

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
