# Peer review of "Association between Emotional Eating, Depressive Symptoms and Laryngopharyngeal Reflux Symptoms in College Students: A Cross-Sectional Study in Hunan"

_nutrients, 2020, doi:10.3390/nu12061595_

Round 1

Reviewer 1 Report

This study investigates a novel relationship between depression, emotional eating, and Laryngopharyngeal reflux in college students. The study was conducted well, but there are several areas that would benefit from additional details.

There were many grammatical errors and typos throughout. This manuscript needs a detailed proof reading and editing before being published.

Section 2.1

Further details about the inclusion/exclusion requirements:

States that no serious physical or mental illness, how was this defined to the participants?

Section 2.2.1

On line 102 it states that “The revised Chinese version of TFEQ-R18V3 has good reliability and validity among college students.” A references should be added to support this.

Need to include a reference for the Food Frequency Questionnaire.

Section 2.2.2

It is stated that height and weight were measured, but earlier it is stated that participants self-reported their height and weight. This should be clarified.

Section 3

The results section should be expanded. It goes into detail on the results detailed in Table 1, but only devotes one sentence to Table 2, and does not even mention Tables 3 or 4.

Reviewer 2 Report

This cross-sectional study aims at better characterize symptoms of the laryngopharyngeal reflux (LPR), in relation with emotional eating and depression in Chinese students.

If the question is of interest owing to the prevalence of LPR and the lack of knowledge concerning the associated symptoms, the paper needs a strong revision before being published in Nutrients.

First, the paper is not well written, it contains lots of mistakes and typographical errors. Some sentences have no sense. Tables are rough and difficult to read.

Second, the choice of the statistical model is questionable and needs to be better explained by the authors. Results section has to be re-written to improve its comprehension and conclusions that are made from analysis clearly formulated.

Introduction

Could the authors specify why emotional eating (EE) which is defined as overeating (line 32) is not considered as an eating disorder?

Before introducing the digestive function, could the authors cite examples of chronicle diseases linked to EE and on the basis of previous publications or arguments they raise the question of the digestive tract? What do they mean by “emotional eating is likely to increase the risk of LPR disease by changing an individual`s dietary patterns”?

Material and methods

Line 102: “The revised Chinese version of TFEQ-R18V3 has good reliability and validity among college students”. Please provide references to support this statement

Line 108: no EE / low EE (no EE:score=0; low EE: score >0 and score < median); and 2) high EE (score ≥ median). As values are not shown in the results sections, it is really difficult to understand cut off placed at 0 or median. A positive score may be > to the median, I suppose? This part must be clarified and results showed in the result section.

Line 118: reference, please use number

Line 119: “To sum up the single indicators to an overall index of PA-related ME (Metabolic equivalent, MET min−1) has been a significant goal of the IPAQ instruments”. What is the meaning of this sentence? Once again raw data are lacking, values of MET in the population would be appreciated.

Line 126: “more and more otolaryngologists”. Idem line 102.

Line 127:  “A total of 45 points. The higher the RSI score, the more severe the LPR symptoms.” Please correct these sentences that are not. The nature of this variable is unclear (raw data are not presented). A better presentation of this score would perhaps allow a better understanding and justification of the logistic regression model used. What was the Cronbach’s alpha of this test?

Line 129: only a subset of data from FFQ was used. Are quantitative data (e.g. calorie consumption) available in this test? If yes, are they not relevant in view of the references cited in the introduction (21-23 and more for GERD)

Line 141 Statistical analysis: the choice of the logistic regression model is questionable. Is the RSI score (used to quantify LPR symptoms) considered as a binary qualitative variable? How were the explanatory variables chosen to construct the model? This part must be detailed and argued. Values of EE score are medians, so what about normality and variance equality of these data (valuable for all scores used in the study), which are prerequisites before performing logistic regression?

Results

Presentation and redaction of the results section must be significantly improved.

Figures showing medians and interquartile of the different scores would be appreciated.

Line 169-173: This part appears as the legend of the Table 1. It is very confusing since it refers to table 2. A true legend for Table 1 is needed to understand P values in the table and how (or why) they differ in the text?

Line 176-182: same remarks for tables 2 and 3. Legend (if it is) is confusing. In the table 2, n=105 and not 115, please correct.

Line 184-191: same remarks for tables 3 and 4. Multivariate logistic regression and Khi2 are mixed in this “legend” leading to confusion.

Discussion

Line 214: “both high-sweet / high-fat diets and high-salt / high-fat diets were positively correlated with LPR”. Data from FFQ test concerning kcal (if available) would be useful to reinforce this point of discussion.

Line 217: could the authors be more precise about “certain eating behaviors”

Line 220: discussion around overeating, emotional eating, binge-eating is of great interest but needs to be clarified. How those behaviors differ and how they can contribute to LPR? Bing-eating is often associated with vomiting (and as a consequence, acid reflux); this point has to be discussed.

Line 226: “the food consumption of emotional eaters was different from others. In our study, individuals with high emotional eating scores consumed more high-energy foods”. More frequently. No quantitative evaluation of kcal intake was performed.

Line 229: “Few studies have reported that the association between food  consumption and LPR disease”. Please correct the grammar of this sentence.

Line 233: please provide a reference?

Line 237: “depression and emotional eating can cause LPR symptoms independently”. Such a causal link is not demonstrated in this study. Please moderate this point of discussion.

Line 242: “decrease their negative emotions effectively”. Can the authors precise how? It would be very interesting because EE is also implied in obesity. It is crucial to resolve or limit EE.

Line 250: “The EE variable was not normally distributed, so, our study used the median(0-100) to represent the emotional eating score. Some studies used the mean to represent the EE score when the sample size was appropriate”. Why the authors make this precision? Please provide the data in the results section. How to reconcile this point (distribution) with the multivariate logistic regression?

Reviewer 3 Report

The study “Association between emotional eating, depressive symptoms and laryngopharyngeal reflux symptoms in college students: A Cross-Sectional Study in Hunan Province” comprises an interesting theme, explored in a cross-sectional design among 1301 young adults from China. However, several alterations are needed in order to improve the quality of the manuscript. The manuscript would benefit of language correction for improving English use and a more formal text construction. Specific comments and suggestions are given below.

Abstract

  1. Objectives: The sentence is redundant. To test the associations, authors need to first evaluate the status, it is expected and doesn’t have to be said in the abstract. I suggest “This study aimed to explore the associations between emotional eating, depression and laryngopharyngeal reflux among college students in Hunan Province.”
  2. Line 21: Please replace “obtained” by “obtain”.
  3. Conclusion: What are the base for defining that 18.6% is a “very common” symptom?
  4. Conclusions need to state the specific answer for your objectives. In this case, authors should say if associations between emotional eating, depression and reflux were found.

Introduction

  1. Line 36. Again, why calling 23.8% as “widespread”? This statement may be too strong.
  2. Line 43. Please replace “,” by “.” In the end of the sentence.

Methods

  1. Lines 74 and 75: Please revise the sentences and edit “The participants of students”.
  2. Line 82. Do not use “didn’t” or similar in the text. Language should not be colloquial. Use “did not”, “do not”, etc.
  3. Line 100. Please explain the variable “major”.
  4. Line 100. Height and weight are not demographic characteristics. They are anthropometric variables. In addition, authors need to mention in some part of the manuscript what is the validity of using self-reported anthropometric information. Please give references to validate this choice.
  5. Lines 101-102: What would be a three-factor eating questionnaire (TFEQ-R18V3) revised by TFEQ-R21?
  6. Line 108. Please define EE on its first appearance in the text, and be sure to clearly define all abbreviations when they are first mentioned in the manuscript.
  7. Line 125: As an example on how to prioritize a more formal writing, I suggest replacing “which was recognized by more and more otolaryngologists” by “which is getting increasingly recognized by otolaryngologists”.
  8. Line 126-127: Another example of the need for editing the text construction: “A total of 45 points.” This is not a sentence. Please revise.
  9. A general suggestion for the text: since authors are describing how the study was done, it is more usual to describe the methods in the past tense, not in the present.
  10. Line 133: Is there any reference that guided authors on defining high intake as ≥3 times/week and low intake as <3 times/week? Please justify.
  11. Line 139: Please add the unit of measurement for BMI (kg/m2).
  12. Line 139: In the first part of the Methods, authors mention that weight and height were self-reported, but then it is said that both measures were assessed by the researchers. Why were self-reported measures collected?

Results

  1. Line 152: Please give the unit of measurement for age (years).
  2. Lines 153-154: Please give the p-value for confirming the finding that BMI was higher in men.
  3. Line 155: If the sentence gives the percentage of obese and overweight subjects separately, it is redundant to also give the percentage of overweight or obese participants.
  4. Line 157-159: The sentence is confusing. It would be enough to say that: “Fewer men than women were emotional eaters (high EE: 48.4% vs. 55.4%; P < 0.001)”.
  5. Lines 159-160: The sentences about the prevalence of depressive symptoms and LPR could be merged into one.
  6. Line 166: Please replace “didn’t” by “did not”.
  7. Lines 169-173: Instead of giving the chi-square value, it would be more valuable to mention the percentages of the variables that are being compared, for each group, and the p-value.
  8. Line 173: Please indicate the direction of the association (higher LPR symptoms?).
  9. Lines 176-178: These descriptive sentences should be given in the beginning of the Results section.
  10. Lines 178-182: Instead of giving the chi-square value, it would be more valuable to mention the percentages of the variables that are being compared, for each group, and the p-value.
  11. Line 185-186: Please replace “with no EE and low EE” by “with no EE or low EE”.
  12. Line 186: Please replace “college students with high EE were likely” by “college students with high EE were more likely”.
  13. Line 190: I do not think the median was a good choice for being the reference group in the analysis of physical activity. What is the clinical message of this analysis? In my opinion, there is no possible clinical message, what is reinforced by the results (both higher and lower than the median physical activity levels provided significant results). I suggest changing the reference group and updating this analysis.
  14. Tables 1 and 2. Please use “P<0.0001”instead of “P<0.000”.
  15. Table 1. Please give the unit of measurement for monthly expenditure.
  16. Tables 1 and 2. All information given in tables must be comprehensible without looking at the text. Please add a footnote to explain the meaning of the abbreviations (BMI, EE, RSI) and to inform which tests were performed to compare the groups.
  17. Table 4. Please inform that the presented values are OR (95% CI).

Discussion

  1. I suggest starting the discussion by briefly resuming the aims of the study and the studied population, to then mention the main findings.
  2. Line 214: Avoid using the term “correlated” to discuss a result of an analysis that was not a correlation test. Prefer using more general terms, as “associated”.
  3. Line 215: Please replace “reduces” by “reducing”.
  4. Lines 236-237: Authors say: “According to our result, depression and emotional eating can cause LPR symptoms independently.” However, it is important not to forget, during all the discussion, that results came from a cross-sectional analysis, which prevents from inferring causality. Authors may indeed propose a possible direction of causality based on biological plausibility and other findings from literature, but caution is needed in this sense.
  5. Line 243: Please replace “decease” by “decrease”.
  6. Line 244: Same comments as in the abstract: what are the base for defining that this is “very common”?
  7. Line 260: Please explain “less or more physical activity”. Again, I believe the choice of median as the reference group made the analysis not clinically relevant. I suggest changing the reference and then updating the necessary parts in the results, tables and discussion.
  8. Lines 269-270: In the sentence: “Medical students and students with high physical activity have a lower percentage of high emotional eating scores” – Are the authors talking about their own results or about other studies in literature? This has to be clear. If mentioning the literature, give references. If talking about the findings of their own study, prefer using the past tense, and give additional elements in the sentence to avoid misunderstandings.
  9. Line 271: Again, I highlight the authors may be leading the readers to misunderstandings. By using the present tense in the sentence: “When they have negative emotions, they still choose a balanced diet.” authors induce the readers to the idea that this would be a causal relationship (what cannot be confirmed by a cross-sectional analysis) or that this is a general statement (in this case, references should be given to support it).
  10. Lines 271-272: If “health education” (and specifically nutritional education) is not even mandatory in elementary or high school, I see no point in calling it “necessary” in university. Maybe authors could reformulate their suggestion of the importance of this type of education earlier in life, to avoid this proportion of college students presenting such conditions when they become adults. But I have difficulties in imaging public policies or national educational programs aiming to include nutritional education in universities.
  11. Lines 273-275: These sentences should had been given in the first paragraph of the discussion, not here.
  12. Line 277-278: Authors mention that “this study did not investigate the effects of other lifestyles on LPR symptoms” – but in fact authors did not investigate the “effects” at all. As previously said, cross-sectional analysis does not allow to infer any causality. Authors tested associations, not effects. Please revise the entire text in the sense of avoiding this type of error.
  13. The paragraph of study limitations does not discuss the cross sectional design of the study. Please include it as a limitation preventing from inferring causality.
  14. Strengths of the study should also be given (example: the sample size).

Conclusions

  1. Line 291: Please replace “and 8.8% of college students” by “and 8.8% among college students”.
  2. The main finding (which answers the main objective) was secondarily mentioned in the conclusions (“depressive symptoms and emotional 291 eating were related to the occurrence of LPR symptoms.” I suggest editing this paragraph in order to highlight the most important findings first.

Round 2

Reviewer 2 Report

The authors have significantly improved their manuscript by taking into account all the remarks that have been made. The reviewer thanks them for this and agrees for publication in Nutrients.

Author Response

Dear reviewer,

We deeply appreciate for your insightful and valuable comments and suggestions. You help us a lot in revising and improving our paper.

Reviewer 3 Report

Authors significantly improved the quality of the manuscript “Association between emotional eating, depressive symptoms and laryngopharyngeal reflux symptoms in college students: A cross-sectional Study in Hunan”. Answers in the rebuttal letter were generally satisfactory, with a few exceptions. Specific comments can be found below.

Point 25:Lines 169-173: Instead of giving the chi-square value, it would be more valuable to mention the percentages of the variables that are being compared, for each group, and the p-value.

Response 25: Thank you for your suggestions. We have revised it to “Individuals reporting high EE (12.4% vs. 3.2%, P<0.001) had higher scores for LPR symptoms than those with no EE or low EE. Students with depressive symptoms (21.5% vs. 5.0%, P< 0.001) showed more LPR symptoms, compared those without depressive symptoms. Both high consumption of sweet and/or fatty foods (24.2% vs. 6.2%, P<0.001) and high consumption of salty fatty foods (24.4% vs. 6.9%, P <0.001) were associated with higher LPR symptoms (Table 2.)”. Line 180-185.

>>>> In the last sentence, please add the group that is being compared (low consumption). In addition, I suggest mentioning the numbers in the end of the sentences: “Individuals reporting high EE had higher scores for LPR symptoms than those with no EE or low EE (12.4% vs. 3.2%, P<0.001). Students with depressive symptoms showed more LPR symptoms, compared those without depressive symptoms (21.5% vs. 5.0%, P< 0.001). Compared to low consumption, both high consumption of sweet and/or fatty foods (24.2% vs. 6.2%, P<0.001) and high consumption of salty fatty foods (24.4% vs. 6.9%, P <0.001) were associated with higher LPR symptoms (Table 2)”.

Point 33: Table 1. Please give the unit of measurement for monthly expenditure.

Response 33: Thank you for your suggestion. We have added the unit of monthly expenditure(RMB). Table 1.

>>>> All abbreviations must be explained, in text or tables. Please add the meaning of RMB in the footnote on the table.

Point 34:Tables 1 and 2. All information given in tables must be comprehensible without looking at the text. Please add a footnote to explain the meaning of the abbreviations (BMI, EE, RSI) and to inform which tests were performed to compare the groups.

Response 34: Thank you for your suggestion. We have added the footnote to explain the meaning of the abbreviations (BMI, EE, RSI) and added this sentence “Chi-square test was performed to compare the groups.” Table 1. and 2.

>>>> "EE" was not explained in footnote of Tables 3 and 4. "OR" was not explained in the footnote of Table 4.

Point 35: Table 4. Please inform that the presented values are OR (95% CI).

Response 35: Thank you. We have added the “OR (95% CI)”,Table 4.

>>>> It seem that only “Crude OR” was added. It is not possible to know that numbers inside parentheses are 95% CI. Please revise, and also explain the meaning of the abbreviations "OR" and "CI" in the footnote. In addition, it is more usual to present the unadjusted values in the first column, to then present adjusted data in the following columns.

Point 48:The paragraph of study limitations does not discuss the cross sectional design of the study. Please include it as a limitation preventing from inferring causality.

Response 48: Thank you for your suggestion. We have revised it “S Second, this was a cross-sectional study that failed to infer causality between EE and, depressive, and LPR symptoms.”, Line 295-296.

>>>> This limitation was not adequately presented. By saying that “this study failed to infer causality”, it seems that it tried to, but did not find causality. This is not the case. Authors were not testing causality at all, because a cross-sectional study is not a design that allows testing causality. Authors were only testing associations. Please revise the sentence. I suggest: “Second, the cross-sectional design of the study does not allow to infer any direction of causality among the studied variables”.
